



# Single-celled bioturbators: benthic foraminifera mediate oxygen penetration and prokaryotic diversity in intertidal sediment

Dewi Langlet[1,2], Florian Mermillod-Blondin[3], Noémie Deldicq[1], Arthur Bauville[4,5], Gwendoline Duong[1], Lara Konecny[3], Mylène Hugoni[6,7], Lionel Denis[1] and Vincent M.P. Bouchet[1]

[1] Univ. Lille, CNRS, IRD, Univ. Littoral Côte d'Opale, UMR 8187, LOG, Laboratoire d'Océanologie et de Géosciences, Station Marine de Wimereux, F-59000, Lille, France.

[2] Evolution, Cell Biology, & Symbiosis Unit, Okinawa Institute of Science and Technology, 1919-1 Tancha, Onna-son, Kunigami-gun, Okinawa, 904-0495, Japan.

[3] Univ. Lyon, Université Claude Bernard Lyon 1, CNRS, ENTPE, UMR 5023 LEHNA, Laboratoire d'Ecologie des Hydrosystèmes Naturels et Anthropisés, F-69622, Villeurbanne, France.

[4] Center for Mathematical Science and Advanced Technology, Japan Agency for Marine-Earth Science and Technology, Yokohama, Japan.

[5] Now at: Axelspace corporation, Tokyo, Japan.

[6] Univ. Lyon, Université Claude Bernard Lyon 1, CNRS, INSA de Lyon, UMR 5240 MAP, Microbiologie, Adaptation et Pathogénie, F-69622, Villeurbanne, France.

[7] Institut Universitaire de France (IUF), 75005 Paris, France.

*Correspondence to*: Dewi Langlet (dewi.langlet@oist.jp) and Vincent M.P. Bouchet (vincent.bouchet@univ-lille.fr)

**Abstract.** Bioturbation processes influence particulate (sediment reworking) and dissolved (bioirrigation) fluxes at the sediment-water interface. Recent works showed that benthic foraminifera largely contribute to sediment reworking in intertidal mudflats; yet their role in bioirrigation processes remains unknown. In a laboratory experiment, we showed that foraminifera motion-behavior increased the oxygen penetration depth and decreased the total organic content. Their activity in the top 5 mm of the sediment also affected prokaryotic community structure. Indeed, in bioturbated sediment, bacterial richness was reduced and sulfate reducing taxa abundance in deeper layers was also reduced, probably inhibited by the larger oxygen penetration depth. Since foraminifera can modify both particulate and dissolved fluxes, their role as bioturbators can no longer be neglected. They are further able to mediate the prokaryotic community, suggesting that they play a major role in the benthic ecosystem functioning and may be the first described single-celled eukaryotic ecosystem engineers.

## 1 Introduction

Intertidal mudflats are among the most productive ecosystems on Earth (Heip et al., 1995). Given their natural features, they are zones of prime importance for organic matter (OM) accumulation (Jickells and Rae, 1997) which can sequester more than 200 gC/m²/year (Chmura et al., 2003). Mudflat sediments usually host intense biological activity and OM is rapidly mineralized (Mayor et al., 2018) via a series of diagenetic reactions from oxygen respiration to methane production (Froelich et al., 1979). In such cohesive environments, dissolved oxygen ($O_2$) is usually available only in the top millimeters of the sediment and transport of solutes is assured by molecular diffusion (Aller, 1988).

Burrow-dwelling macro-invertebrates (organisms larger than 500 µm) greatly influence intertidal mudflats functioning through bioturbation (Meysman et al., 2006) – a process which combines sediment reworking (i.e. transport of particles) and





bioirrigation (i.e. transport of water and solutes, see review in Kristensen et al. (2012). The effects of bioturbation by macro-invertebrates on the benthic ecosystem functioning is mediated by complex interactions with meiofaunal organisms (organisms smaller than 500 µm; Piot et al. 2014; Lacoste et al. 2018; Schratzberger and Ingels 2018). Indeed, meiofauna

may also contribute significantly to sediment reworking (Bradshaw et al., 2006) and bioirrigation (Cullen, 1973; Aller and Aller, 1992). Noticeably, meiofauna was reported to improve sediment oxygenation and sulfide removal (Bonaglia et al., 2020), to affect nitrogen cycle by stimulating nitrate reduction (Prast et al., 2007; Bonaglia et al., 2014) and to enhance OM mineralization (Rysgaard et al., 2000; Nascimento et al., 2012). Meiofaunal bioturbation can further lead to changes in the abundances of all and specific groups of bacteria in sediments (Prast et al., 2007; Lacoste et al., 2018; Bonaglia et al., 2020)

but these studies did not evaluate its effect on the whole bacterial and archaeal community structures. Bioturbation by macro-invertebrates may significantly impact bacterial community structure by modifying biogeochemical gradients and by modifying the availability and quality of OM (e.g., mucus production) in sediments (Papaspyrou et al., 2006; Cuny et al., 2007). For example, Laverock et al. (2010) demonstrated that bacterial communities from irrigated burrows of the ghost shrimp (*Upogebia deltaura* and *Callianassa subterranea*) were more diverse than bacterial communities from non-

bioturbated sediments. In this context, it can be expected that bioirrigation by meiofauna would similarly increase oxygen availability in sediments, hence favouring aerobic prokaryotes over strictly anaerobic species in sediments.

In spite of their role in benthic ecosystem functioning (Moodley et al., 2000; Geslin et al., 2011), the role of foraminifera as bioturbators remains a fairly untapped question, with only a few pioneer studies looking at how their displacements may affect sediment reworking process (Severin et al., 1982; Hemleben and Kitazato, 1995; Groß, 2000). Noticeably, their ability

to move in the sediment column affects the surface sediment cohesiveness (Cedhagen et al., 2021) and contributes to the horizontal and vertical transport of sediment particles (Groß, 2002; Deldicq et al., 2020, 2021, 2023). Consequently, foraminifera are assumed to affect sediment porosity and allow for "good sediment ventilation" (Hemleben and Kitazato, 1995; Groß, 2002). Supporting this assumption, foraminiferal activity was shown to affect dissolved cadmium concentrations in the pore-water and overlaying water (Green and Chandler, 1994) suggesting that foraminifera influence the

water and solutes exchanges at the sediment-water interface. However, studies based on two-dimensional oxygen measurements did not report a positive effect of foraminifera on dissolved oxygen concentrations in sediments as their aerobic respiration produced a decrease of oxygen penetration depth in foraminiferal burrows (Oguri et al., 2006; Heinz and Geslin, 2012).

In this context, it appears critical to further describe the role of foraminifera in bioirrigation processes and quantify their

contribution to solute fluxes at the sediment-water interface. To do so, the impact of foraminiferal displacements in the sediment matrix was assessed on 1) the oxygen vertical distribution in homogenized sediment, 2) the subsequent oxygen fluxes at the sediment-water interface, 3) the resulting influences on OM content (total organic carbon and total nitrogen) and 4) the prokaryotic (archaea and bacteria) community structure to ultimately determine their role in bioirrigation processes, OM mineralization and the microbenthic communities.




## 2 Material and methods

### 2.1 Sediment and living foraminifera collection

Surface sediment (top 10 mm) from Authie Bay (Northern France, English Channel, 50°22'20"N, 1° 35'45"E) was collected in January 2018 and kept frozen in the dark at -20°C to kill any potential bioturbators before being used in the experimental
cores.

Living foraminifera were extracted from surface sediment (top 10 mm) collected in the Boulogne-sur-Mer harbor (50°43'04"N 1°34'26"E) in November 2019. Only active individuals (i.e. leaving a displacement track on a thin layer of sediment) were selected for the experiment.

### 2.2 Experimental design

A total of 17 cores (45 mm height and 10 x 10 mm square section, Figure 1A) were filled with homogenized thawed Authie Bay sediment, placed in an air-bubbled 7 L aquarium (closed system filled with 35PSU unfiltered English Channel seawater), and left for 14 days prior to adding foraminifera to give enough equilibration time to establish redox fronts and microbial processes in the sediment column. The experiment was carried out for 85 days in the dark (with a photosynthetic active radiation < 0.7 µmol photon/m²/s; SA-190 quantum sensor, LI-COR) in a temperature-controlled room (at 18 ± 1°C).
Oxygen microprofiles were realized in control cores (n = 6, without any foraminifera) and cores with foraminifera (n = 6, abundance = 30 indiv/cm²). Foraminiferal species composition (78% *Haynesina germanica*, 10% *Ammonia tepida*, 8% *Quinqueloculina seminulum* and 4% *Cribroelphidium excavatum* per core) and abundance selected for the experiment were chosen based on their natural densities and species composition in local mudflats (Francescangeli et al., 2020). From these 12 cores, 3 control cores and 3 cores with foraminifera were randomly selected at the end of the experiment to evaluate the
influence of foraminifera on organic matter (OM) content and microbial community structures at two sediment depths (0-5 mm and 5-10 mm). The remaining three cores with foraminifera were used to determine the foraminiferal survival rate. Eight cores containing no foraminifera were dedicated solely to microporosity measurements at the beginning (n = 4) and at the end of the experiment (n = 4).

### 2.3 Foraminifera survival

At the end of the experiment, 3 cores with foraminifera were placed in a 1 µmol/L CellHunt Green CMFDA solution (5-chloromethylfluorescein diacetate, Setareh Biotech) for 24 hours, fixed with 70% ethanol and sieved over a 125 µm mesh (Choquel et al., 2021; Langlet et al., 2013). Foraminifera exhibiting a bright fluorescence under an epifluorescence stereomicroscope (Olympus SZX16 with a fluorescent light source Olympus KL1600pE -300) at 492 nm excitation and 517 nm emission wavelength (Langlet et al., 2014) were picked and identified to determine foraminiferal survival rate.





### 2.4 Organic matter measurements

Total organic carbon (TOC) and total nitrogen (TN) contents of sediment samples were measured following the capsule method (Brodie et al., 2011). They were determined by high-temperature combustion of pre-acidified (HCl, 2N) dry samples (60°C, 48 h) and subsequent measurement of $CO_2$ and $N_2$ by thermal conductometry using an elemental analyzer (FlashEA, Thermo Electron Corporation).

### 2.5 Microporosity measurements

At the beginning and at the end of the experiment, 4 sediment cores were frozen at -20°C and sliced with a razor blade from 0 to 10 mm depth with a 1 mm vertical resolution to measure water content. For each slice of sediment, we measured on a precision microbalance (Sartorius R160P) the humid ($m_h$) and dry ($m_d$) masses (before and after drying at 40°C for 48 hours) to determine water mass ($m_w$ such as $m_w = m_h - m_d$) and calculate the sediment microporosity ($\Phi$) with $\varrho_w = 1.035$ and $\varrho_s = 2.65$ the density of water and sediment respectively (Berner, 1980). Microporosity vertical distribution was modeled following an exponential decrease with depth (Supp. Figure 2). To estimate microporosity at each sampling time, we assumed that it was decreasing linearly with time.

### 2.6 Pore-water dissolved oxygen distribution

### 2.6.1 Sampling strategy

Each core dedicated to oxygen profiling was subdivided into 5 zones (Fig. 1B) to ensure that microprofiling was not realized twice in the same area. At each measurement time (from 1 day before adding foraminifera to 85 days after introduction of the living foraminifera), 2 cores containing foraminifera and 2 control cores were analyzed with 3 oxygen microprofiles realized per core (Fig. 1C). All measuring zones were selected randomly to minimize any potential effect of microtopography and core-specific response (Supp. Table 2).

### 2.6.2 Oxygen microprofiling

At each sampling time, a 50-µm tip diameter Clark type microelectrode (Revsbech, 1989) (Unisense, Denmark) was 2-points calibrated using the overlying water in the air-bubbled aquarium as 100% saturation reference and the signal at 10 mm depth in the experiment sediment as anoxic reference. Oxygen concentration at 100% saturation in 18°C and 35PSU sea water was 239.7 µmol/L. The microsensor was placed on a motorized micromanipulator (Unisense, Denmark) and vertical profiles were realized from about 2 mm above the sediment-water interface down to the anoxic zone of the sediment with a 150-µm vertical resolution. Three microprofiles were realized in each selected zone and the distance between two replicate profiles ranged from about 1 to 2 mm.



### 2.6.3 Oxygen profile interpretation

The oxygen penetration depth (OPD) was selected as the shallowest point with a dissolved oxygen concentration lower than

1 µmol/L (Bonaglia et al., 2020).

We computed diffusive oxygen uptake (DOU) following (Berg et al., 1998), eq. 1-10. We minimized the cost function, which includes data from the three replicates, using the L-BFGS-B algorithm (Byrd et al., 1995) with bounds to ensure that production remained negative. Berg et al. (1998) employed the stepwise regression algorithm that results in piecewise constant "production zones" (their eq. 11) to limit the complexity of the model. Instead, we regularized the total variation

(i.e., the sum of the absolute first-order derivative) using the elastic net algorithm (Rudin et al., 1992). Like the number of zones in Berg et al. (1998), the regularization intensity is a hyperparameter that controls the complexity (i.e., smoothness) of the optimized profile. We provide the algorithm, data and Jupyter notebook to reproduce our analysis (see supplementary material).

We imposed nil oxygen concentration and nil DOU in the sediment at the bottom of the calculation zone (Bonaglia et al.,

2014). The diffusion coefficient ($Ds$) was calculated using the microporosity ($\Phi$) measurements ($Ds = D_0 * \Phi^2$; Ullman and Aller 1982) and a $D_0$ coefficient of 1.854 $10^{-5}$ cm$^2$/s (oxygen diffusion coefficient at 18°C and 35 PSU).

### 2.7 Prokaryotic diversity

For each sample, DNA was extracted from 0.25 g of wet sediment using the ZymoBIOMICS™ DNA Miniprep Kit (Zymo Research), according to the manufacturer's instructions. The quantity and the quality of extracted DNA were quantified and

controlled using PicoGreen and a capillary electrophoresis (QIAxcel), respectively. V3-V5 hypervariable regions of the 16S gene were amplified to target bacterial community and archaeal community, and to evaluate the respective abundances of archaea and bacteria in sediments. Amplifications were done using the following primer pairs: 357F_ILMN (5'-CCTACGGGAGGCAGCAG-3') and 926R_ILMN (5'-CCGYCAATTYMTTTRAGTTT-3') for bacteria, 519F_ILMN (5'-CAGCMGCCGCGGTAA-3') and 915R_ILMN (5'- GTGCTCCCCCGCCAATTCCT-3') for archaea, and 515F_ILMN (5'-

GTGYCAGCMGCCGCGGTA-3') and 909R_ILMN (5'- CCCCGYCAATTCMTTTRAGT-3') for relative abundances of archaea and bacteria. First PCR (PCR 1) was performed with 35 cycles at 50°C for bacteria and at 58°C for archaea and relative abundances. Each PCR1 was performed in a 25 µL reaction volume, using "5x HOT BIOAmp ® BlendMaster Mix" DNA Polymerase, 2 µL of DNA template, 0.24 µmol/L reverse and forward primers, MgCl$_2$ at 12.5 mmol/L, Bovine Serum Albumin at 20 mg/mL, "10x GC rich Enhancer ", and nuclease-free water. Thermal cycles were as follows: 95°C for 3 min

(95°C for 30s, 55°C for 30s, 72°C for 1 min) 25 times, and 72°C for 5 min. The PCR was replicated three times for the 12 samples and 2 controls (extraction and PCR controls) for each couple of primers. Amplification replicates were then pooled and purified using Agencourt AMPure XP beads. A second PCR (using PCR1 as DNA template) with 15 cycles for bacteria and archaea and 10 cycles for relative abundances was performed for sample indexing (indexes+P5/P7). PCR2 products were also purified with AMPure beads. Then, DNA was quantified using the Quantifluor dsDNA kit (ThermoFisher). All



samples were pooled in equimolar proportions and sequenced on an Illumina MiSeq platform with 5% PhiX (Flow Cell V3, Paired-End 2 * 300 bp) by Biofidal (Vaulx-en-Velin, France, http://www.biofidal.com).

Bioinformatic processing of the merged 2x300 bp paired-end reads followed sequential steps: 1) dereplication and filtering (keeping only 300 to 500 bp –long reads containing a valid mismatch-free tag and no ambiguous base), 2) clustering into operational taxonomic units (OTUs) with SWARM (Mahé et al., 2014) (two-step-procedure: local clustering threshold d=1

and then d=3), 3) removal of chimera, 4) removal of OTUs detected in only one out of three replicates from same condition, 5) abundance normalization (by rarefaction, i.e. subsampling at 33,885 reads per sample for bacteria, 33,834 reads per sample for archaea, and 15,645 reads per sample for respective abundances, to correct for variability in sequencing depths among samples) and 6) taxonomic affiliation against the 16S SILVA database release 138 (Quast et al., 2013), based on NCBI blastn+ (Altschul et al., 1990) and allowing for multiple affiliation. These different steps were performed with FROGS

(Find Rapidly OTUs with Galaxy Solution; Escudié et al. 2018) on the Galaxy web platform (Afgan et al., 2018) of the Pôle Rhône-Alpes de Bioinformatique. The OTU abundance tables, and taxonomic assignments produced at this stage were then analyzed using the vegan R package (Oksanen et al., 2020) to calculate alpha diversity indices (OTU richness and Shannon index).

## 2.8 Statistical analysis

Since oxygen microprofiles were measured several times in a same core, we chose to analyze the effect of foraminiferal bioturbation using linear mixed-effects models (Pinheiro and Bates, 2000) with "core" as a random effect in all models. Oxygen penetration depth (OPD) and dissolved oxygen uptake (DOU) were set as response variables while experiment time, treatment (control or with foraminifera) and time-treatment interaction were selected as fixed effects. Preliminary analysis showed a shift in oxygen conditions between days 36 and 55, hence modeling was performed on data acquired from 0-36

days and 55-85 days separately. Due the peculiar shape of the oxygen distribution profiles, data acquired on Day 5 (zones J4, K2, D2 and F2) both in controls and cores with foraminifera were removed from the analysis (see supplementary figure 1).

The influence of sediment layer and treatment on sedimentary bacterial (or archaeal) community structure was visualized using a non-metric multidimensional scaling (NMDS) performed with data of OTU abundances obtained from the different cores. Differences in bacterial (or archaeal) community structures between sediment layers and treatments were tested using

permutational multivariate analyses of variance (PERMANOVA; Anderson 2001). Statistical tests were based on 999 permutations of the Bray-Curtis matrix.

To determine whether the experiment affected strictly anaerobic micro-organisms, supplementary analyses were performed on bacterial taxa involved in sulfate reduction and archaeal taxa involved in methane production. Three sulfate-reducing bacterial orders (Desulfatobacterales, Desulfovibrionales and Synthrophobacterales) were selected based on the literature

(Wasmund et al., 2017). Their relative abundances (proportion of reads) in bacterial communities were determined for each sample. The same procedure was applied on the relative proportion of methanogens from three classes of archaea





(Methanobacteriales, Methanosarcinales and Methanomicrobiales). Relative abundances of sulfate-reducers and methanogens were logit-transformed to normalize their distributions.

The influence of sediment depth (0-5 mm and 5-10 mm) and treatment (control or with foraminifera) on TOC and TN
content, bacterial and archaeal diversity indexes (OTU richness and Shannon diversity), sulfate-reducing bacterias and methanogenic archaea were tested using a 2-way ANOVA (ANOVA2) with sediment layer and treatment as main effects. For all variables, the normality and the homoscedasticity of the residues were tested using Shapiro-Wilk's test and Levene's test, respectively. When these assumptions were not met, data were log-transformed before statistical analyses using 2-way ANOVA. Data analysis was carried out in R v.3.5.3 using nlme, ade4 and vegan packages (Pinheiro and Bates, 2000; Dray
and Dufour, 2007; R Core Team, 2019; Oksanen et al., 2020).

## 3 Results

### 3.1 Foraminiferal activity observations

Non-quantitative observations showed sediment displacement at the sediment surface as well as burrow formation on the sides of sediment cores down to about 7 mm depth. Newly formed burrows were frequently observed during the first 3
weeks of experiment, but no new burrows were found after 3 weeks. Investigation of the CellHunt Green-labeled sediment at the end of the experiment showed 19, 22 and 26 living foraminifera corresponding to a survival rate of 63, 73 and 87% in the 3 tested cores.

### 3.2 Sediment organic carbon and total nitrogen content

At the end of the experiment, total organic carbon (TOC) content ranged from 1.4 to 1.7% and total nitrogen (TN) ranged
from 0.21 to 0.27% (Figure 2). In the top sediment layer (0-5mm) TOC was significantly lower in the cores with foraminifera than in the control cores (1.4% ± 0.05 standard deviation and 1.6% ± 0.07, respectively) while no significant differences were observed in the 5-10 mm layer (2-way ANOVA, interaction "treatment * sediment layer", $F_{(1,8)}=10.4$ and $p < 0.05$). Similarly, TN was significantly lower in the top layer of the cores with foraminifera than in the control cores (0.2% ± 0.01 and 0.3% ± 0.01, respectively) while no effect of foraminifera was observed in the deeper sediment layers ($F_{(1,8)}=8.9$
and $p < 0.05$).

### 3.3 Oxygen distribution in the sediment

Replicated dissolved oxygen microprofiles were homogeneous within each sampling zones and modeled oxygen profiles used for dissolved oxygen uptake (DOU) estimates showed good fit with the measured data ($R^2 > 0.97$; Supplementary figure 1).
During the first 36 days of the experiment, oxygen penetration depth (OPD) ranged from 2.1 to 3.6 mm in the control cores and from 2.4 to 4.2 mm in the cores with foraminifera (Fig. 3A). Linear mixed effect models showed a significant effect of



the Time * Treatment interaction (Table 1) such as OPD was stable with time in the controls and increased by about 0.7 mm in sediment with foraminifera over the course of the first 36 days of experiment.

After 55 days, OPD ranged from 3.6 to 4.5 mm (Figure 3A) and did not show any significant differences between the cores with foraminifera and the control cores (Table 1).

DOU ranged from 0.011 to 0.033 nmol/cm²/s (Figure 3B) and were stable in the control cores while they significantly decreased from 0.025 to 0.011 nmol/cm²/s during the first 36 days of the experiment in the cores with foraminifera (Table 1). At 36 days of experiment, DOU in the bioturbated cores were in average 0.013 nmol/cm²/s lower than in the control cores. After 55 days, DOU ranged from 0.006 to 0.014 nmol/cm²/s and did not significantly differ between treatments (Table 1).

### 3.4 Prokaryote community structures

Bacterial communities dominated prokaryotic communities with more than 97% of reads corresponding to bacterial OTUs and less than 3% of reads related to archaeal OTUs. The relative abundance of bacterial OTUs in prokaryotic communities significantly increased with depth with 97% of bacteria in the 0-5 mm sediment layer and 99.5% in the 5-10 mm sediment layer (ANOVA2, depth effect, $F_{(1,8)}=67.3$, $p<0.001$). Furthermore, bacterial richness was positively correlated to TOC ($R^2 = 0.46$, $p<0.01$).

The most abundant phyla in the sediment were Proteobacteria, Chloroflexi, Bacteroidetes, and Actinobacteria (Fig. 4A). The NMDS analysis and PERMANOVA tests showed significant differences in bacterial community structures between depths (Figure 5B, sediment layer effect, PERMANOVA, $F_{(1,10)}=13.1$, $p<0.005$). Indeed, phylum-level analyses showed that the relative abundance of Bacteroidetes in bacterial community increased with depth whereas the opposite pattern was observed for Proteobacteria (Fig. 4A). Although the presence of foraminifera did not significantly influence the bacterial community structures (PERMANOVA, foraminifera effect, $F_{(1,10)}=0.53$, $p>0.6$), the foraminiferal activity significantly reduced bacterial richness in the top sediment layer (Fig. 4C, ANOVA2, interaction "sediment layer * foraminifera treatment", $F_{(1,8)}=6.3$, $p<0.05$). This effect of foraminifera on bacterial OTU numbers was not detected on Shannon diversity considering the relative abundance of each bacterial OTU (ANOVA2, $F_{(1,8)}<0.9$ and $p>0.05$ for both foraminifera treatment and "foraminifera treatment * sediment layer" interaction). It is also worth noting that bacterial diversity significantly decreased with depth for both control and bioturbated cores (Fig. 4C, ANOVA2, sediment layer effect, $F_{(1,8)}=106$ and $p<0.0001$).

Specific analyses performed on the main sulfate-reducing orders of bacteria (Desulfatobacterales, Desulfovibrionales and Synthrophobacterales) showed that the relative abundances (% of reads) of these three orders within bacterial communities increased with depth (Fig. 5A, ANOVA2, sediment layer effect, $F_{(1,8)}=54$ and $p<0.0001$). The presence of foraminifera co-occurs with a 20% reduction of the relative abundance of sulfate-reducing orders in the deepest layer of sediment (ANOVA2, interaction "sediment layer * foraminifera effect", $F_{(1,8)}=6.5$ and $p<0.05$).

Archaeal communities were dominated by Thaumarchaeota in the 0-5 mm depth layer and by Woesearchaeota in the 5-10 mm depth layer (Figure 6A). The pattern observed with depth for Thaumarchaeota was due to the genus *Candidatus Nitrosopumilus* which represented more than 80% of reads of the archaeal community sampled in the 0-5 mm depth layer



whereas it corresponded to less than 15% of reads from the 5-10 mm depth layer. Consequently, NMDS and PERMANOVA tests showed a clear influence of sediment depth on the structure of the archaeal community (Figure 6B, PERMANOVA, $F_{(1,11)}=38.3$, $p<0.005$). This effect was likely due to significant increase in archaeal richness and diversity between sampled sediment layers (ANOVA2, sediment layer effect, $F_{(1,8)}>100$ and $p<0.0001$ for archaeal richness and Shannon diversity). In comparison, no significant effect of the treatment was detected on archaeal community structure (PERMANOVA,

$F_{(1,11)}=0.1815$, $p>0.82$), archaeal richness (ANOVA2, foraminifera effect, $F_{(1,8)}=1.1$, $p>0.32$) and archaeal diversity (ANOVA2, foraminifera effect, $F_{(1,8)}=1.6$, $p>0.23$). Taxa specific analyses on relative abundances of methanogenic archaea in communities (Methanobacteriales, Methanosarcinales and Methanomicrobiales) also revealed no significant influence of the presence of foraminifera (ANOVA2, foraminifera effect, $F_{(1,8)}= 1.8$, $p>0.21$) whereas the proportion of methanogens in communities increased with depth (Figure 5B, ANOVA2, layer sediment effect, $F_{(1,8)}= 90.1$, $p<0.0001$).

## 4 Discussion

### 4.1 Oxygen and organic matter as main determinants of microbial communities in undisturbed sediments

The decreasing vertical gradients of dissolved oxygen measured in sediments usually determine the vertical distribution of microbial communities (Fenchel and Finlay, 2008). In the control cores of our experiment, non-metric dimensional scaling (NMDS) results clearly demonstrated that the bacterial and archaeal communities were structured by the sediment depth and

the associated oxygen availability in pore water. For example, the archaeal genus *Candidatus Nitrosopumilus*, involved in nitrification process, showed a preferential distribution in the 0-5 mm sediment layer because this genus needs oxygen to oxidize $NH_4^+$ into $NO_2^-$ and $NO_3^-$ (Walker et al., 2010). In addition, oxygen penetration depth ranged from 2 to 5 mm in undisturbed cores and strict-anaerobic microorganisms like sulfate-reducing bacteria and methanogenic archaea were more represented in the communities found in the anoxic 5-10 mm sediment layer than in the shallowest sediment layer (0-5 mm).

Without organic matter (OM) addition during the experiment, we also observed that the total organic carbon (TOC) content was lower in the upper sediment layer than in the deep layer likely due to the positive influence of oxygen availability on the mineralization of OM in sediments. Indeed, the aerobic mineralization of sedimentary OM is known to be faster than anaerobic mineralization, irrespective of the degree of lability of OM (Kristensen et al., 1995). The vertical distribution of dissolved oxygen in sediments was thus determinant on OM dynamics and the structure of microbial communities. In turn,

the vertical gradient of TOC and TN in sediments generated by OM mineralization could also shape the bacterial community. For example, the lower representation of phylum Bacteroidetes – which are abundant in nutrient-rich aquatic environments (Landa et al., 2013) - in the top sediment layer compared with the bottom layer could be due to the low OM measured at the end the experiment.

Overall, in undisturbed sediment, both oxygen and OM availability were the main parameters structuring microbial

communities in the present experiment. In such conditions, we can expect that if foraminiferal activities modify these two determinants, they would in turn modulate the microbial compartment.





Finally, we observed an increase of oxygen penetration depth (OPD) and a decrease of diffusive oxygen uptake (DOU) between 36 and 55 days of experiment in our control sediment. Similar observations were made previously in sediment without meiofauna between 5 and 14 days of experiment (Bonaglia et al., 2020). Although the kinetic is different (likely due

to the different nature of the sediment used in the two experiments), we may hypothesize that the decrease of available OM throughout the experiment led to non-linear changes in OPD and DOU in the control cores.

### 4.2 Foraminiferal motion activity

In our experiment, benthic foraminifera built up burrows down to 7 mm in the sediment. Although these burrows were not as deep as cm-long burrows previously reported on miliolid and some deep-sea species (Severin et al., 1982; Groß, 2002; Heinz

and Geslin, 2012), they were in the same order of magnitude as known for the coastal species *Ammonia beccarii* (Green and Chandler, 1994). These shallow burrows confirm that the intertidal foraminiferal species used in the present experiment prefer oxygenated microhabitats (Bouchet et al., 2009; Cesbron et al., 2016). However, foraminifera could burrow 2 mm deeper than the maximal oxygen penetration depth measured in the experimental cores. Although foraminiferal mobility is known to be inhibited by low oxygen concentration (Maire et al., 2016), it seems that during our experiment, the community

dominated by *H. germanica* remained active even below the oxygen penetration depth, suggesting that their burrows might provide enough dissolved oxygen to sustain their activity.

Despite this tolerance to low oxygen concentration, observations showed that foraminifera mainly created their burrows during the first three weeks of the experiment and no new burrow could be observed during the period lasting from 40 to 90 days. This contrasts with previous reports suggesting that frequently fed deep-sea foraminifera can continuously generate

new burrows over the course of several years (Hemleben and Kitazato, 1995). The difference could come from behavioral differences between deep sea foraminifera and the coastal species used in our experiment or due to the lack of added food in our setup which might have starved the foraminifera hence limited their long-term activity.

Despite this potential limitation of foraminiferal activity by fresh OM, the TOC content measured at the end of the experiment in sediments (from 1.4 to 1.7%) was in the same order of magnitude as contents usually reported from sediments

of the Authie Bay and Boulogne-Sur-Mer harbor (ranging from 1 to 1.7%; Francescangeli et al. 2020). Although their reduced activity at the end of the experiment may likely be due to the absence of fresh OM input, foraminiferal survival remained high with on average 75% of the individuals found alive after 85 days of experiment, stressing that the experimental conditions were close to those observed in the field.

### 4.3 Foraminiferal bioturbation stimulates aerobic organic matter mineralization

Foraminiferal activity in the first month of experiment resulted in a significant increase of OPD with an average difference of about 0.7 mm between the bioturbated and control cores on day 36. It therefore suggests that benthic foraminiferal burrowing activity increased the volume of oxygenated sediment by about 20% which is in the same order of magnitude as previously reported in other meiofaunal organisms (Bonaglia et al., 2020). In both foraminifera (this study, day 36) and



meiofauna (Bonaglia et al., 2014, 2020), the OPD enhancement led to a decrease of DOU in bioturbated cores suggesting
that foraminifera affect dissolved fluxes in a similar way as meiofaunal ostracods, nematodes, copepods and oligochaetes.

Nevertheless, macro-invertebrates and meiofaunal organisms seem to have different impacts on benthic oxygen fluxes.
Bioturbating macro-invertebrates tend to increase both the DOU (Forster and Graf, 1995; Volkenborn et al., 2007; Lagauzère
et al., 2009) and the total oxygen uptake (TOU, Kristensen 1985; Pelegrí and Blackburn 1994; Michaud et al. 2005; Politi et
al. 2021). In contrast, meiofaunal bioturbation leads to a decrease in DOU (this study, Bonaglia et al. 2014, 2020) and an
increase in TOU (Bonaglia et al., 2014). In the freshwater environment, bioirrigation by chironomid larvae increased DOU
in organic-matter poor sediment whereas the same bioturbation activity decreased DOU in organic-matter rich sediment
(Stief and de Beer, 2002) suggesting that OM availability and benthic microbes respiration mitigates the effect of
bioturbators on diffusive oxygen fluxes. In our experiment, the decrease of TOC in cores with foraminifera suggests an
increase in OM mineralization. Hence, the decrease in DOU would likely be a consequence of the reduced OM availability
in bioturbated cores.

In previous work, the reduced DOU was interpreted as an increase of meiofaunal predatory pressure on their bacterial preys
leading to a decrease in the population of aerobic prokaryotes (Bonaglia et al., 2014). In our study, bacterial richness was
positively correlated to TOC suggesting that the low bacterial richness in sediment layers bioturbated by foraminifera was
due to low OM content rather than a top-down control by predation. A similar mechanism was described in freshwater
sediments with tubificid worms which reduced the quantity and the quality of the sedimentary OM by stimulating OM
mineralization, leading, in turns, to a decrease in bacterial richness and diversity (Cariou et al., 2021).

As the availability of fresh OM had a significant control on bacterial community structures in marine sediments (Deng et al.,
2020), foraminifera most likely reduced the quality (consuming the most labile fraction of OM) and the diversity of the OM
in sediments by stimulating OM mineralization (i.e., total organic carbon loss) during the three months of the present
experiment. Consequently, the availability and diversity of OM was more limiting in bioturbated than in non-bioturbated
sediments, hence reducing the ability of multiple bacterial taxa to coexist (increased competition with the reduction of
trophic niches; Langenheder et al. 2010; Šimek et al. 2014). Such reduction of the number of trophic niches available in the
sedimentary column would have then decreased the bacterial richness. Nevertheless, this effect was not observed on
Shannon bacterial diversity because the reduction of OM associated with foraminifera activities probably affected low-
abundant (rare) OTUs which have a lower influence on Shannon diversity index than on bacterial richness (e.g., Haegeman
et al. 2013). It is also worth noting that the collection of samples for microbial communities was done after 85 days of
experiment when the effect of foraminifera on dissolved oxygen gradient was not significant. In these conditions, we can
expect that microbial changes were less marked at this date than after one month of experiment when foraminifera had the
strongest effect of oxygen concentrations in sediments. As already mentioned for sulfate-reducing bacteria and
methanogenic archaea, the availability of dissolved oxygen was recognized as a main structuring factor of microbial
community structure and biogeochemical process in marine sediments (Kristensen and Holmer, 2001; Bertics and Ziebis,





2009). Thus, future experiments should measure the dynamics of microbial communities during experiments to evaluate the potential time-dependent effects of foraminiferal bioturbation on the microbial compartment.

**4.4 Foraminifera modulate anaerobic diagenetic processes**

In our study, benthic foraminifera improved the pore-water oxygenation, and their burrows might also affect a series of diagenetic processes. Indeed, coastal foraminifera are known to accumulate large amounts of nitrate in their cells (Geslin et al., 2014; Langlet et al., 2014; LeKieffre et al., 2022) and deep-sea foraminifera can reduce nitrate and greatly contribute to benthic denitrification (Langlet et al., 2020; Choquel et al., 2021). Our results suggest that foraminiferal bioturbation also affected the benthic nitrogen cycle since lower total nitrogen (TN) content were measured in sediments bioturbated by
foraminifera in comparison with non-bioturbated sediments. Similar decreases in TN have been reported in sediments bioturbated by macro-invertebrates (Shen et al., 2017; Cariou et al., 2021) and were likely due to increased mineralization of OM associated with a denitrification process. Several bioturbating meiofaunal organisms (including rotifers, polychaetes, oligochaetes, crustaceans, ciliates and nematods) were also shown to affect benthic nitrogen cycle by enhancing microbial denitrification (Rysgaard et al., 2000; Prast et al., 2007; Bonaglia et al., 2014). Although not quantified in this experiment,
we can expect that foraminiferal bioturbation might affect microbial denitrification in a similar way as other meiofaunal organisms. Thus, further experiments using $^{15}$N-nitrate tracing methods (Bonaglia et al., 2019) will be necessary to determine whether foraminifera contribute to benthic nitrogen cycle via enhancing the denitrifying activity of microorganisms by bioturbation.

Furthermore, the enhancement of oxygen penetration depth by meiofaunal bioturbation can accelerate sulfide removal
(Bonaglia et al., 2020). Ventilation of ghost shrimp burrows was also reported to increase sulfate reduction in oxygenated micro-niches (Bertics and Ziebis, 2010). In addition, bioturbation can control the community composition sulfate-reducing bacteria (as shown in meiofauna Bonaglia et al. 2020), and the abundance of active sulfate-reducing bacteria (as shown in macro-invertebrates (Mermillod-Blondin et al., 2004). In our experiment, the low relative abundance of sulfate-reducing bacterial OTUs in the deepest layer (5-10 mm) of bioturbated cores suggests that foraminiferal bioirrigation might similarly
inhibit sulfate-reduction in the sediment. Foraminifera are known to be sensitive to free-sulfide (Bouchet et al., 2007; Richirt et al., 2020) so the oxygenation of their burrows likely provide sulfide-free microhabitat in deeper sediment layers.

Finally, our analysis on the proportion of methanogenic archaeal groups in the community did not support the hypothesis that foraminiferal bioturbation activity influenced methanogenic process in sediments. This corroborates previous experiments showing no effect of bioturbating meiofauna on methane fluxes (Bonaglia et al., 2014). Methanogenesis usually
occurs in deeper sediment layers in organic-matter rich sediments (Froelich et al., 1979). Methane production is likely minimal in the top centimeter of the sediments used in the present experiment as indicated by the low relative abundance of methanogenic archaea (<2% of all the archaea population). Further experiments using deep-dwelling foraminiferal species and organic-matter rich sediment would be of great interest to evaluate the potential role of these organisms in the benthic methane cycle.

**4.5 Foraminifera as ecosystem engineers**

Our results clearly show that foraminifera, at densities commonly reported in coastal environments, affect oxygen distribution and fluxes in the sediment via their burrowing activity. If previous studies showed that foraminifera rework sediment (Groß, 2002; Deldicq et al., 2021), the present study takes our knowledge a step further in showing that they can also perform bioirrigation; hence, foraminifera should now be considered as bioturbators. To the best of our knowledge, there is no report of other single-celled eukaryotes having such a broad impact on the benthic ecosystem functioning suggesting that foraminifera might be the first described single-celled ecosystem engineers.

Foraminiferal vertical distribution pattern is known to be affected by macrofaunal bioturbation (Bouchet et al., 2009; Thibault de Chanvalon et al., 2015; Maire et al., 2016) and meiofaunal bioturbation processes are deeply interconnected with macrofaunal organisms (Nascimento et al., 2012; Bonaglia et al., 2014; Lacoste et al., 2018). To fully discuss the role of foraminiferal bioturbation on benthic ecosystem functioning it now appears necessary to further study their interactions with other benthic compartments such as meio- and macrofauna.

**Data and materials availability:**

All other data are available in the main text or the supplementary materials.

**Author contributions:**

Conceptualisation: DL, LD, VMPB

Investigation: DL, FMB, ND, AB, GD, LK, MH

Visualization: DL, FMB

Supervision: LD, VMB

Writing-original draft: DL, FMB

Writing-review & editing: DL, FMB, ND, AB, LD, VMPB

**Competing interests**

The authors declare no competing interests.

**Acknowledgements**

We thank Pr. Edouard Metzger and Dr. Florian Cesbron for their helpful comments on oxygen profile interpretation and Jean Charles Pavard for his assistance with foraminiferal extraction in preparation of the experiment. The authors thank the



Région Hauts-de-France, and the Ministère de l'Enseignement Supérieur et de la Recherche (CPER Climibio), and the European Fund for Regional Economic Development for their financial support. DL was supported by the Région Hauts-de-France STaRS fellowship COFFEE and the Japan Society for the Promotion of Science.

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





**Figure 1A:** schematic representation of experimented control cores and cores with foraminifera (side view) at the beginning (Day 0) and the end of the experiment (Day 85) with cores sampled for measurements of microporosity (grey), prokaryotic diversity and sediment TOC and TN (brown), foraminifera survival (green) and O₂ micro profiling (blue). **B,** location of the 5 microprofiling zones (top view of the cores) and **C,** picture of the cores placed in the aquaria during oxygen microprofiling.



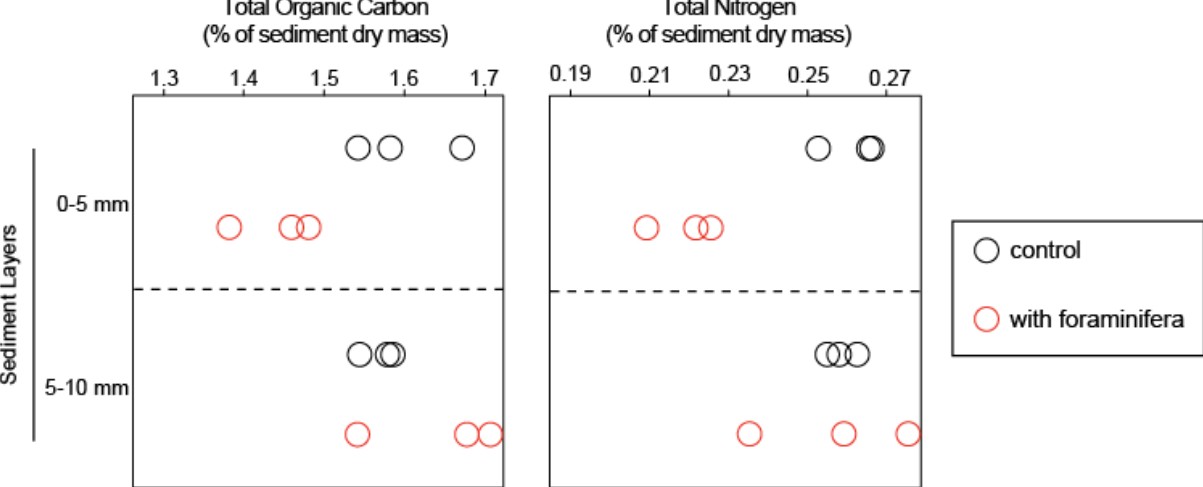

**Figure 2: Percentages of total organic carbon and total nitrogen per sediment dry mass for control (black open circles) and foraminifera (red open circles) treatments in two sediment layers sampled at the end of the experiment (85 days) in 3 replicate cores.**




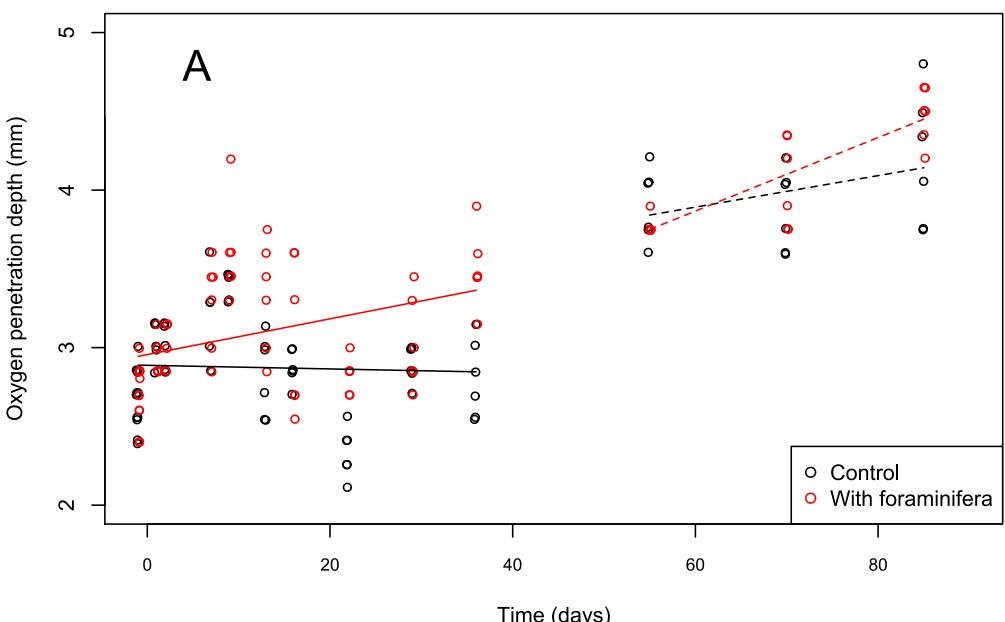

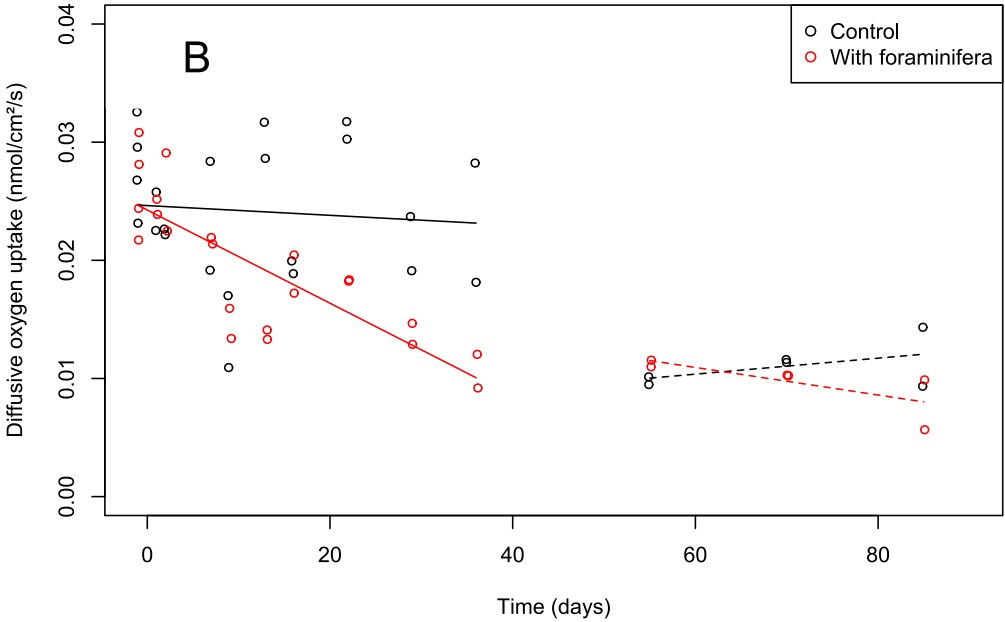

**Figure 3: Changes of the oxygen penetration depth (A) and dissolved oxygen uptake (B) with sampling time in the control (black) and bioturbated cores (red). To visually differentiate the otherwise identical values, a small amount of noise was added to the data** 660 **(with a jitter factor 0.5 on both x and y axes). Lines were plotted based on the linear models estimates (see Supplementary Table 1)**



and drawn as full or dashed line when the Time x Treatment variable was significant or insignificant (at a 0.05 threshold) respectively.

**Figure 4A)** Bacterial community structure (relative abundance) in control (C) and bioturbated (F) cores in two sediment layers. Community structure is represented at the phylum level. Only phyla representing at least 1% of the reads in at least one sample are represented. **B)** Non-metric multidimensional scaling of the bacterial communities in control ("C" and black open circles) and bioturbated ("F" and red open circles) cores in the 0-5 mm ("H1: labels) and 5-10 mm ("H2" labels) sediment layers. **C)** Richness



**and diversity of bacterial communities in the different sediment layers in 3 control (black open circles) and 3 bioturbated (red open circles) cores.**

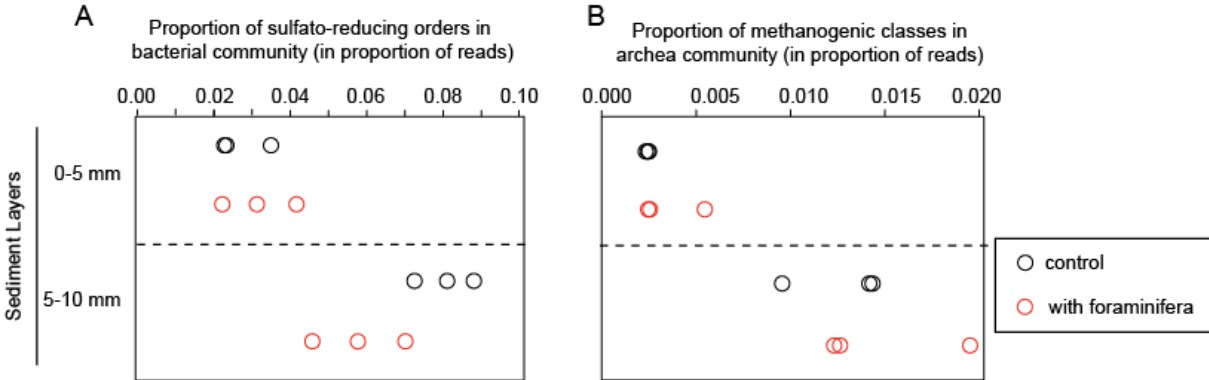

**Figure 5A) Proportion of reads in bacterial community corresponding to three orders involved in sulfate-reduction process (Desulfatobacterales, Desulfovibrionales and Synthrophobacterales) and B) Proportion of reads in archaeal community corresponding to three classes of methanogens (Methanobacteriales, Methanosarcinales and Methanomicrobiales). Values are shown in the different sediment layers for 3 control (black open circles) and 3 bioturbated (red open circles) cores.**







**Figure 6A) Archaeal community structure (relative abundance) in control (C) and bioturbated (F) cores in two sediment layers. Community structure is represented at the phylum level. Only phyla representing at least 1% of the reads in at least one sample are represented. B) Non-metric multidimensional scaling of the archaeal communities in control ("C" and black open circles) and bioturbated ("F" and red open circles) cores in the 0-5 mm ("H1" labels) and 5-10 mm ("H2" labels) sediment layers. C) Richness**





**and diversity of archaeal communities in the different sediment layers in 3 control (black open circles) and 3 bioturbated (red open circles) cores.**

**Table 1: Statistics of the effect of the experimental treatment on the sediment oxygen parameters. Results of the statistical analysis (linear mixed-effect models with "core" as random effect) for all dataset (time period day 0-36 and 55-85) and fixed effect variables. Variables showing a significant effect on the response variable (p<0.05) are shown as bold characters and with stars (\* when p<0.05, \*\* when p<0.001 and \*\*\* when p<0.0001).**

| Dataset | Response variable | Fixed effect variable | numDF | denDF | F | p-value | Sign. |
|---|---|---|---|---|---|---|---|
| Day 0 - 36 | Oxygen penetration depth | **Intercept** | **1** | **118** | **3855.6** | **<0.0001** | *** |
| | | Time | 1 | 118 | 3.6 | 0.061 | |
| | | **Treatment** | **1** | **10** | **5.5** | **0.041** | * |
| | | **Time * Treatment** | **1** | **118** | **5.4** | **0.022** | * |
| | Diffusive oxygen uptake | **Intercept** | **1** | **30** | **894.0** | **<0.0001** | *** |
| | | **Time** | **1** | **30** | **12.8** | **0.001** | ** |
| | | **Treatment** | **1** | **10** | **10.1** | **0.010** | * |
| | | **Time * Treatment** | **1** | **30** | **8.4** | **0.007** | * |
| Day 55 - 85 | Oxygen penetration depth | **Intercept** | **1** | **24** | **2567.3** | **<0.0001** | *** |
| | | **Time** | **1** | **8** | **6.5** | **0.034** | * |
| | | Treatment | 1 | 8 | 0.5 | 0.517 | |
| | | Time * Treatment | 1 | 8 | 1.0 | 0.336 | |
| | Diffusive oxygen uptake | **Intercept** | **1** | **8** | **451.3** | **<0.0001** | * |
| | | Time | 1 | 8 | 0.4 | 0.557 | |
| | | Treatment | 1 | 8 | 1.7 | 0.229 | |
| | | Time * Treatment | 1 | 8 | 5.3 | 0.050 | |