# Peer review of "Single-celled bioturbators: benthic foraminifera mediate oxygen penetration and prokaryotic diversity in intertidal sediment"

_EGUsphere, 2023_

## Author Comment (AC1)

**Referee 1**

In general, the manuscript is well written, and the topic is highly interesting. However, prior to publication I have some questions and remarks that should be clarified prior to publication. For more details, please see below.

**Methods section**

Experimental set up -section needs some clarification. Currently it is not clear how the sediment was homogenised? How was it ensured? The process is also likely to have had an impact on the porosity and water content of the sediment and hence the sediment structure may be different to what it is at the field situation. Could this have implications for the results and how the situation is in nature?

> Response: Sediment was defrosted at room temperature in a glass beaker, stirred with a glass rod and pipetted in the measurement cores. Cores were left to settle for a few days to let the geochemical gradients to establish. Figure S2 shows the vertical microporosity distribution with typical exponential decrease of water content with depth. Differences between the T0 and T85 showed that the sediment porosity decreased during the experiment following the natural process of compaction. This suggests that despite its initial homogenization, the vertical shape of sediment porosity was similar to those that can be observed in natural sediments (after a resuspension event, for instance).

> Furthermore, it is worth reminding that both the control sediment and sediment with foraminifera were treated in the same way allowing for proper quantification of the effect of foraminifera on benthic fluxes despite potential experimental biases related to the nature of the sediment.

What size of foraminifera were picked and introduced into the experiment, also >125 um as sieved at the end of the experiment?

> Response: Yes, foraminifera > 125µm were picked and introduced in the experiment. This information will be added to the revised manuscript.

It is also not clear how often and how many times in total the oxygen profiling was conducted. Based on the Figure 3 profiles were conducted during 13 days? Was every control core (n=6) and foram core (n=6) measured on these days as a triplicate? Or were some cores measured more than others? In the section 2.6.2 it is stated that each core was divided into 5 areas to avoid multiple measurements in a same spot, implying that each core was measured max 5 times. The frequency and number or measurements per core should be clarified. I would also be interested to know if the authors have considered the impact of O2 profiling on the oxygen penetration depth in the sediment. In the end a considerable number of profiles were made, each producing a small vertical burrow down to several mm into sediment.

> Response: Detailed sampling plan is presented in Table S2.

> According to manufacturer information, each oxygen profile has a significant influence on an area corresponding to twice the tip diameter (50 µm), so here perturbations might be significant at a scale close to 100 µm, while the distance between replicate profiles were more than 1mm. Moreover, it is possible to make successive profiles in the hole made by the microsensor without affecting profiles. We chose to make 3 separate wholes separated by at least 1 mm to limit the impact of successive profile acquisitions in each microprofiling zone (see Figure 1B). In consequence we assume that successive microprofiles in similar microprofiling zones had a minimal effect on the oxygen penetration depth. Since both the control and with foraminifera sediment were treated in the same way, the bias introduced by microprofiling is similar in both types of cores allowing for proper quantification of the effect of foraminifera on pore-water oxygen microdistribution.

Please clarify in section 2.7 how the sampling was conducted, and from which depth intervals samples were taken. I assume that somehow the core was sliced, or was sediment scooped straight from the core? If latter, how was mixing of the sediment avoided, and contamination? Please clarify the text.

> Response: At the end of the experiment, the core was frozen at -20 degrees. Few days later the sediment was pushed out of the core and sliced in 5mm intervals using a sterile razor blade. Information about sample processing will be added to the section 2.7.

**Results section**

Section 3.2 did you measure the TOC and N-content at the start of the experiment? What was it at the start prior to incubations and how did it differ from the control cores at the end of the experiment? The freezing of the sediment used for the experiments would have introduced some fresh OM to the sediment, consisting of meiofauna etc. Foraminiferal are known the feed on various prey (e.g. Chronopoulou et al. 2019  DOI: 10.3389/fmicb.2019.01169), so wondering how much of the OM was processed by foraminifera, and if the difference in the Corg content at the end is due to foraminiferal grazing or enhanced OM-processing by bacteria due to bioirrigation?

> Response: Unfortunately, we did not measure the TOC and TN at the start of the experiment. But given the homogenization procedure it is fair to assume that these values did not differ between the control and cores with foraminifera.

> Based on *Haynesina germanica* oxygen respiration rates (981 pmol/ind/day; Deldicq et al., 2021; https://www.nature.com/articles/s41598-021-83311-z) and the foraminiferal carbon mineralization conversion proposed in Geslin et al. 2011 (https://doi.org/10.1016/j.jembe.2010.10.011) we can estimate that 30 *H. germanica* (the main species in our experiment) can mineralize **9 mgC/cm$^2$/month**. Our TOC measurements suggest that there is a 1.4 mgC/cm$^2$ (assuming a 1.7g/cm$^3$ sediment density) difference in the top 5 mm between control and cores with forams at the end of the experiment. This would mean that foraminifera would have grazed about **0.45 mgC/cm$^2$/month**. This suggest that foraminiferal aerobic mineralization is larger than the differences in TOC observed at the end of the experiment. Yet, these calculations are based on a lot of assumptions (such as the extrapolation of minutes-long respiration to 3 month long experimental measurements) so we think that it is complicated to seriously consider reporting these numbers in the manuscript.

Section 3.3. It seems that the O2 penetration depth was relatively shallow at 22 days. Could some of the differences in the OPD-measurements be due to different numbers of O2 profiles made in the cores? Or do authors have an idea why ODP was shallow at 22 days?

> Response: We realized the same number of profiles at all sampling times. If there are any differences, they are not due to changes in the sampling effort. We did not observe any changes in the experimental conditions at that time so we do not have any hypothesis to explain lower OPD values at that peculiar sampling time. Even if the sediments were homogenized with great care before the beginning of the experiment, we cannot rule out the fact that the shallow OPD at 22 days might be linked to spatial heterogeneity (higher OM accumulation?).

Section 3.4. It is a shame that the authors did not take a molecular sample at the start of the incubation, to observed how to prokaryote community developed from T=0 to T=85. If such a sample was taken, it would be very valuable to analyse it and include the results here.

> Response: We agree with the reviewer. Unfortunately, we did not take any samples at the beginning of the experiment, and we did not conserve sediment cores from the beginning of the experiment.

**Discussion section**

Section 4.1 line 275. Without the Corg and N-measurements for the T=0, the authors cannot for certain confirm that that the Corg declined during the experiment.

> Response: In this sentence we do not discuss the temporal dynamics of OM but rather focus on the differences measured between surface and deeper sediment layers at the end of the experiment.

Section 4.1 and elsewhere: I suggest that authors do not call control cores undisturbed but always as control cores, as the control cores were disturbed during the experiment with O2 microprofiling. It would be interesting, for authors to discuss the influence of profiling on the results. I.e. if some control cores were measured more than others, is there a difference in Corg, N, or OPD or O2 uptake etc?

> Response: the word undisturbed will be replaced by control all throughout the manuscript. There were no differences in the number of profiles realized in the control cores so our data will not be able to help us discuss this question. Each profile being realized at least 1mm apart from the previous profile, and the sensor being very thin (50 μm at the tip), it is unlikely that the hole left by the previous profile affects the following profile OPD. The plot bellow shows the distribution of the measured OPD in profiles 1, 2 and 3 where we could not observe any significant increase of the OPD between replicates (ANOVA, df=1, 166, F=0.006, p=0.94). Suggesting that the holes made in profiles 1 and 2 did not affect the oxygen distribution of profile 3.

[Figure]

Line 293, and 303: how were the burrows measured? Is it possible there were longer burrows inside the sediment cores that are not visible from observing the core from outside? And also that burrows were made inside the cores after 40 days.

> Response: burrow depth was assessed visually. We unfortunately do not have any good quality images that allow for more accurate measurements yet this value is in the same order of magnitude as those reported in Deldicq et al. 2023 (https://doi.org/10.1098/rspb.2023.0193) using similar sediment and the most abundant species in our study (*Haynesina germanica*). It is possible that deeper burrows were not visible inside the cores and that some were created after 40 days. We only did not observe them. Discussion will be modified to clarify this information.

Line 332-334 (and elsewhere related to bacterial richness estimate from OTUs) Bacterial richness estimation is based on number of OTUs. This, however, is somewhat problematic as OTUs do not directly translate to number of taxa (as some OTUs can be from same species). The more recent concept of an amplicon sequence variant (or ASV) could be a more appropriate means of estimating number of taxa present/bacterial richness. Authors could consider looking into ASVs, or at least they should be careful when interpreting bacterial richness based on OTUs and explain the limitations. Especially here, as the Shannon index is not showing the same trends.

> Response: We do not agree with the reviewer. Indeed, ASVs in their definition are used to decipher genetic diversity of sequence variants. In contrast, OTUs reflect the specific diversity, i.e. richness. In this work, we clearly aimed at investigating microbial species, that is why we used OTUs.
>
> From microbiome data, alpha diversity classical indexes considered richness (evaluated like we did through the number of OTUs or even through Chao1 that is an extrapolation of the number of OTUs) and diversity (in which account both abundance and composition, typically through Shannon index). Here, it is not surprising that richness and diversity were different, the explanation is given line 344 "Nevertheless, this effect was not observed on Shannon bacterial diversity because the reduction of OM associated with foraminifera activities probably affected low-abundant (rare) OTUs which have a lower influence on Shannon diversity index than on bacterial richness (e.g. Haegeman et al. 2013)."

Line 358-360. Regarding the influence of foraminifera on sediment N-content. It is more likely that the sedimentary N-content here is a reflection of OM degradation than related to foraminiferal denitrification. During degradation of OM, typically N-containing molecules (e.g. amino acids) are broken down preferentially, hence causing a shift in C/N ratio of OM left in the sediment (e.g. Schneider et al 2003, https://doi.org/10.1029/2002GB001871)

> Response: As explained in the lines 361-367 we also think that the effect of sediment N-content is due to microbial OM degradation. We will modify this paragraph to lift the confusion between microbial OM degradation and foraminiferal denitrification.

**Referee 2**

**General comments:**

First of all, please keep in mind that the following comments come from a place of deep respect and recognition of your work and the energy required to build, perform, and analyze the data of work like this one. The aim of these comments is to improve my understanding of the manuscript with the hope that more people will be able to understand the importance of the presented results.

The paper by Langlet et al. untitled Single-celled bioturbators: benthic foraminifera mediate oxygen penetration and prokaryotic diversity in intertidal sediment is a valuable piece of work documenting for the first time a single-celled ecosystem engineer. This research highlights the importance of foraminifera in the functioning of ecosystems through their influence on sediment irrigation.

The MS is, overall, well written and understandable. However, since English is not my first language, I did not pay attention to the spelling and grammar.

I think the method need some work to clarify some points detailed in my specific comments. I have some issues with the models build for statistical analyses and the consideration of temporal and spatial dependency between sample. Also, no details are given on the precisions and limit of quantification of the analyses. This should be assessing each time especially in this kind of study where differences between treatment are very small. Without this information one could not tell if the differences detected by the author fall within the error of the analytical method. One of my major concerns is the shift in oxygen dynamics over the experimental period. The authors did not give explanation except for the reference of Bonaglia et al. 2020 where sediment characteristics were not comparable (very high OM content). Finally, I think the discussion is well written but not clearly sustained by the results given the information at my disposal. To conclude, I think the manuscript need major revisions to be considered for publication in biogeosciences.

**Introduction**

L36-37: sensu Kristensen et al. 2012 bioturbation is an umbrella term encompassing sediment reworking and burrow ventilation which causes bioirrigation.

> Response: Definition of bioturbation will be modified accordingly.

L50-51: you already mentioned that meiofauna increase oxygen availability (L41). This sentence oversimplifies bioturbation process. Yes, it overall increase oxygen availability but also increase sediment heterogeneity by creating microenvironment and this may be the main cause of more diverse bacterial communities in bioturbated sediment.

> Response: This sentence will be rephrased to include this precision about sediment heterogeneity.

**Material and methods**

L82: Is 14 days enough time to reach a steady state?

> Response: We did not observe any major shift in geochemical gradients in control cores in the first 36 days of the experiment (as indicated by the stable oxygen penetration depth, Figure 3) suggesting that the system was at a steady state when the organisms were introduced after 14 days of equilibration.

L88-91: Did you assess OM content at the beginning of the experiment. Did you assume the OM content was similar between the sediment core at the beginning? That is probably true but worth mentioning.

> Response: We did not assess the OM content at the beginning of the experiment. We assumed to sediment homogenization process allowed for similar OM content in the different cores. This will be added to the revised version of the manuscript.

L96: Why 125 µm? it could be great to mention your definition of meiofauna and justify this mesh size in your context.

> Response: This mesh size was the one used for living foraminiferal collection at the beginning of the experiment (information will be added to section 2.1) and is widely used in foraminiferal ecology research since smaller-size foraminifera can be difficult to identify (REFs).

L111-112: What is your rational for the decreasing of microporosity over time? the sediment core did not reach a steady state after 14 days of acclimation? Why perform microporosity only on sediment without forams? In the introduction you mentioned the impact of meiofauna bioturbation on particle mixing. Do you assume that the particle mixing by meiofauna is negligible?

Response: Decrease of microporosity over time is due to sediment compaction. We believe that the core reached a steady biogeochemical state as suggested by the stable oxygen microdistribution in the control cores (before 36 days), yet, sediment remained to be influenced by compaction. We agree that it would have been interesting to measure microporosity in cores with foraminifera, yet the cores with foraminifera were limited to 6 and we had to choose how to use them. We decided to dedicate 3 replicates to the quantification of foraminiferal survival and 3 replicates for the organic matter and prokaryotic community analyses.

L116: How many samplings time during your experiment? 5? When?

Response: Each core was sampled 5 times during the experiment. Complete sampling plan is indicated in Table S2.

L115-119: This is hard to follow even with the help of Fig. 1. The microprofiles were performed in the same 2 cores over the experiment? Above you mentioned n = 6.

Response: Section 2.6.1 will be rephrased to explain more clearly the sampling strategy.

L177: there is a dependence of data between two experiment time. Therefore, this factor should be random to account for this time dependency.

Response: Does the reviewer mean that time should be added to the model as both a random effect and fixed effect (option A) or that time should be removed to the fixed effect to be included only as random effect (option B)?

Option A: fixed = Time*Treatment, random=Time+Core

Option B: fixed = Treatment, random=Time+Core

L178-180: Please expand on this shift in oxygen condition. Why does it change? Is this a result from a contamination of the control cores? An experimental artefact? This is a major concern since most of your data were collected after this shift.

Response: The shift in oxygen condition is assumed to be linked with a non-linear consumption of the organic matter in the experimental setup. The final part of the section 4.1 will be enriched to further discussed the nature and potential origin of this non-linear change.

L187-193: Do these bacterial orders perform only sulfate-reducing processes? Same question for the methanogens. Also, I have no data on the sediment characteristics, but do you think sulfate-reducing processes and methanogenesis occur in the first cm of the sediment column? With the OM concentration given below it would be very surprising. So why a focus on these processes?

Response: like most marine prokaryotes, these order are indeed flexible and present multiple metabolic activites (Dorries et al. 2016; https://doi.org/10.1002/pmic.201600041 ). Yet taken together these order can be considered as a proxy for sulfate reduction (Wasmund et al. 2017; https://doi.org/10.1111/1758-2229.12538 ). In the revised discussion, we will precise that these orders could have more diverse functions that only sulfate reduction.
We also have no direct measurements of sulfate-reduction nor methanogenesis and agree that it is unlikely that these processes are dominant in the top cm of the sediment as suggested by the low relative abundances of these proxy-taxa. We look at these processes since the prokaryotic dataset allowed to test it and we found interesting to report that there is a potential interaction between foraminifera and sulfate-reducing prokaryotes. We also found interesting to report negative results regarding methanogenic prokaryotes.

L194-196: Did you account for the spatial dependency between sample collected at 0-5mm and 5-10 mm in the same core?

Response: We did not account for the spatial dependency between samples collected in the different levels. In the revised manuscript we will modify the analysis to use linear mixed effect models to account for the "core" effect as random effect. Using this approach, sediment depth (Horizon) and its interaction with Treatment have a significant effect on both TOC and TN (Table R1)

| Dependant variable | Independent variable | numDF | denDF | F-value | p-value |
|---|---|---|---|---|---|
| TOC | (Intercept) | 1 | 4 | 4415.622 | <.0001 |
| | Horizon | 1 | 4 | 19.69 | 0.0114 |
| | Treatment | 1 | 4 | 0.787 | 0.4252 |
| | Horizon:Treatment | 1 | 4 | 35.927 | 0.0039 |
| TN | (Intercept) | 1 | 4 | 4415.622 | <.0001 |
| | Horizon | 1 | 4 | 19.69 | 0.0114 |
| | Treatment | 1 | 4 | 0.787 | 0.4252 |
| | Horizon:Treatment | 1 | 4 | 35.927 | 0.0039 |

Table R1 – Results of the linear mixed effect models testing the effects of sediment depth, treatment and their interaction on TOC and TN.

L198-199: Did the assumption met after log-transformation of the data? Did you try using glm instead of lm models?

Response: This sentence will be rephrased to specify that the assumptions were met after the transformation of the data. We did not try to use glm instead of lm since conditions to use lm were met after data transformation.

**Results**

L210-213: Ok this is significant but does a difference of 0.2 % is significant for forams? Also, what are the (i) precision, (ii) limit of detection and (iii) limit of quantification of your analysis?

L213-215: Idem

Response: In this study we do not aim at determining if the OM matter changes affect the foraminiferal behavior, so determining if the 0.2% difference is significant for the forams is out of the scope of this manuscript.

Average difference between 2 sub-samples was 0.06% and 0.007% for TOC and TN respectively. Assuming that these differences were not due to sediment heterogeneity measurement precision represent about 37 and 17% of the measured TOC and TN differences between control and cores with foraminifera in the 0-5mm depth interval. Method section 2.4 will be modified to precise that sediment samples were measured in two subsamples and give values of the differences between two subsamples as a precision proxy.

L224-225: Again, I am really surprised by this shift in oxygen dynamic during the experiment. With the information at my disposal, I have major concern about the good execution of the experiment. If there is a natural shift in oxygen dynamic, you could use segmented analysis or multi change point analysis to correctly model the relationship and assess the time at which the change occurs.

Response: Thank you for the suggestion. Analysis using the *segmented* R package on the OPD in the control cores suggests that 4 breakpoints maximize the OPD~time model $R^2$ and minimize the

model AIC and BIC (Figure R2).

[Figure]

Figure R2 – r.squared, AIC and BIC of the segmented linear models on the core OPD with 1 to 5 breakpoints settings.

The segmented analysis predicts breakpoints at 1, 9, 22 and 55 days. Using those breakpoints, we ran 5 linear models (since we now have only 3 to 4 sampling time per time intervals we believe it would not be longer possible to use linear mixed effect models with "Core" as random effect) testing the effects of Time, Treatment and their interactions in the 5 time intervals selected by the segmented analysis. Analysis of variance (Table R2) showed that Treatment only had a significant effect on the OPD in the 9-22 days and 22-55 days time intervals. Such as the OPD is about 850 µm higher in the cores with foraminifera than in the control cores at 22 days (Figure R3).

| Time interval | Variable | Df | Sum Sq | Mean Sq | F value | p-value | Signifi |
|---|---|---|---|---|---|---|---|
| "-1 to 1 day" | **Time** | 1 | **740139** | **740139** | **25,35** | **0,00002** | * |
| | Treatment | 1 | 278 | 278 | 0,01 | 0,92 | |
| | Time * Treatment | 1 | 40139 | 40139 | 1,38 | 0,25 | |
| "1 to 9 days" | **Time** | 1 | **1650932** | **1650932** | **37,23** | **0,0000002** | * |
| | Treatment | 1 | 46875 | 46875 | 1,06 | 0,31 | |
| | Time * Treatment | 1 | 174312 | 174312 | 3,93 | 0,054 | |
| "9 to 22 days" | **Time** | 1 | **4920750** | **4920750** | **67,66** | **0,0000000** | * |
| | **Treatment** | 1 | **1522969** | **1522969** | **20,94** | **0,00004** | * |
| | Time * Treatment | 1 | 60750 | 60750 | 0,84 | 0,37 | |
| "22 to 55 days" | **Time** | 1 | **9941160** | **9941160** | **178,52** | **0,0000000** | * |
| | **Treatment** | 1 | **1080000** | **1080000** | **19,39** | **0,0001** | * |
| | **Time * Treatment** | 1 | **461764** | **461764** | **8,29** | **0,01** | * |
| "55 to 85 days" | **Time** | 1 | **1500000** | **1500000** | **21,56** | **0,0001** | * |
| | Treatment | 1 | 105625 | 105625 | 1,52 | 0,23 | |
| | Time * Treatment | 1 | 240000 | 240000 | 3,45 | 0,07 | |

Table R2 – analysis of variance of the linear models in the 5 time intervals.

[Figure]

Figure R3 – OPD depending on time in control (black) and cores with foraminifera (red). Whole line indicates time intervals where Treatment had a significant effect while dashed line indicates time intervals without a significant effect of treatment on the OPD.

If the reviewers agree with this new procedure, we will modify the Figure 3, Table 1 in the manuscript and update the methods and results sections. These new results do not change the main results and interpretations of this paper showing a significant effect of the presence of foraminifera on the oxygen penetration depth.

L226-229: Please consider changing the unit of the DOUs to avoid using decimals. This is, in my opinion, hard to read.

Response: Unit will be changed accordingly.

L234-235: the correlation test is not described in the methods section.

Response: Correlation test will be added to the methods section.

L239: sensu stricto you did not test for an increase of Bacteroidetes with depth, you compared two groups.

Response: Sentence will be modified accordingly.

L242: Ok, there is a significant difference but when we look at the figure, we can see that one sample with forams is in the range of the samples without forams. Do you think here you can detect a significant difference because of heterogeneity of variance? BTW, could you mention which data sets were transformed?

> Response: In this test, like others, the homoscedasticity condition was respected. We will also add to the methods that the list of datasets that were log-transformed.

L249-251: I am not sure to understand what you mean here.

> Response: Sentence will be rephrased to precise that the relative abundance of sulfate-reducing orders in the 5-10mm depth interval is significantly different in cores with foraminifera than in control cores. And that this difference is such as there is a 20% reduction of sulfate-reducing prokaryotes in cores with foraminifera.

**Discussion**

L281-283: Here you refer to the .2 % (TOC) and .05 % (TN) differences? From Fig. 2 the variability in foram treatment within the 0-5mm layer is .1 % vs .2 % within the 5-10mm layer (TOC). In my opinion this is a too small difference to sustain your rational.

> Response: in this section of the discussion, we are only discussing the results observed in the control cores and not discussing the potential effect of foraminiferal bioturbation. We will clarify this all throughout the section 4.1 by specifying that we are talking about the control cores.

L290-291: Without assessing OM at the beginning of the experiment you cannot know if there was a decrease of OM content. Furthermore, in the study of Bonaglia et al., 2020, the TOC was 20 times higher than in your work which can explain the presence of H2S in the first cm of the sediment column.

> Response: Sentence will be rephrased to precise that although not supported by measurements at the beginning of the experiment we hypothesize that the OM availability decreased during the experiment. We will also precise that the TOC was larger in Bonaglia et al. 2020's study.

L332-334: OM account for 46 % of the variability in you model. Could predatory pressure account for the remaining 54 %? Did you test it?

> Response: We believe that it would be complicated to estimate foraminiferal predatory pressure from our data. We know that in average one specimen of *Haynesina germanica* (the dominant species in our experiment) can ingest about 15000 bacteria per hour (Mojtahid et al. 2011, https://doi.org/10.1016/j.jembe.2011.01.011) which would represent $10^7$ bacteria per population of 30 *H. germanica* in a day. Yet, these values are difficult to compare to our richness estimates based on number of OTUs. If the reviewer has any suggestions on how to estimate the foraminiferal predatory pressure from our dataset, we would be happy to hear about it.

L340: by diversity of OM you mean diversity in the quality of OM? How a diversity could be limiting?

> Response: Here we meant "quality" instead of "diversity". This will be corrected in the next version of the manuscript.

L370: Do you mean "reduced" instead of "oxygenated"?

> Response: Yes we meant "reduced". This also will be corrected in the next version of the manuscript.

L376: Did you measure iron and iron oxide? it could be linked to the presence of iron oxydes and the formation of AVS. Forams could also enhance the iron cycling of course.

> Response: Although not quantified, we observed some iron oxides deposits on the wall of all polystyrene cores. So yes, iron oxides were present in the system but since we have no measure of it, we think that it would be complicated for us to discuss the potential interactions between iron and sulfur cycles and how foraminiferal bioturbation modulates it.

L386-389: consider reformulate these sentences. We knew forams were bioturbators mixing sediment particles but now, with your work we know they also alter porewater distribution. I do not think organism should perform sediment reworking AND burrow ventilation to be considered as bioturbators. Some bioturbators do not burrow but crawl at the sediment surface.

> Response: in Kristensen et al. 2012, bioturbation is defined as *"transport processes [...] including both particle reworking and burrow ventilation"*. With our understanding of the meaning of "both", we think that only organisms that perform sediment reworking and burrow ventilation should be considered as bioturbators.

L391: you should add a reference for the definition of EES.

> Response: we will refer to Jones et al. 1994 who defines ecosystem engineers as organisms which "directly or indirectly modulate the availability of resources to other species, by causing physical state changes in biotic or abiotic materials. In so doing they modify, maintain and/or create habitats".

**Additional point**

The authors of the manuscript "Single-celled bioturbators: benthic foraminifera mediate oxygen penetration and prokaryotic diversity in intertidal sediment" thank the two reviewers for their reviews and constructive comments. In addition to the response to their specific comments, we would like to propose to add to the discussion section the following figure (Figure R4) that illustrates our vision on the hypothetical foraminiferal bio-irrigation mechanisms.

[Figure]

Figure R4 – Hypothetical foraminiferal bio-irrigation mechanisms such as foraminiferal burrows enhance the OPD leading in turns to a decrease in DOU and Sulfate reducing prokaryotes. Blue arrows denote the hypothetical effect of foraminiferal bioirrigation on the total oxygen uptake leading to the decrease of sediment TOC and TN and in turns prokaryotic richness.

---

## Author Response (AR1)

Evolution, Cell Biology, and Symbiosis Unit
Dr. Dewi Langlet
Onna, Okinawa, 14th August 2023

1919-1 Tancha, Onna-son,
Okinawa, 904-0495 Japan
Phone. +81-98-966-8711
http://www.oist.jp

Object: resubmission of the manuscript "Foraminifera: single-celled bioturbators"

Dear editor,

Please find hereafter our revision of the manuscript entitled "Single-celled bioturbators: benthic foraminifera mediate oxygen penetration and prokaryotic diversity in intertidal sediment" by Langlet et al..

All changes were made following our previous point-by-point response to the reviewers (https://doi.org/10.5194/egusphere-2023-705-AC1 and https://doi.org/10.5194/egusphere-2023-705-AC2). You can find the revised manuscript as well as the track-changes version.

Sincerely,

Dr. Dewi Langlet, on behalf of the authors

---

## Author Response (AR2)

Evolution, Cell Biology, and Symbiosis Unit
Dr. Dewi Langlet
Onna, Okinawa, 23rd August 2023

**OIST**

OKINAWA INSTITUTE
OF SCIENCE AND TECHNOLOGY
GRADUATE UNIVERSITY

1919-1 Tancha, Onna-son,
Okinawa, 904-0495 Japan
Phone. +81-98-966-8711
http://www.oist.jp

Object: resubmission of the manuscript "Foraminifera: single-celled bioturbators"

Dear editor,

we corrected the manuscript following all suggestions

Sincerely,

Dr. Dewi Langlet, on behalf of the authors